# Genome-Wide Analysis of Abscisic Acid Biosynthesis, Catabolism, and Signaling in Sorghum Bicolor under Saline-Alkali Stress

**DOI:** 10.3390/biom9120823

**Published:** 2019-12-03

**Authors:** Siqi Ma, Lin Lv, Chen Meng, Chao Zhou, Jie Fu, Xiangling Shen, Chengsheng Zhang, Yiqiang Li

**Affiliations:** 1Marine Agriculture Research Center, Tobacco Research Institute of Chinese Academy of Agricultural Sciences, Qingdao 266101, China; masiqi@caas.cn (S.M.); LL19940203@163.com (L.L.); mengchen01@caas.cn (C.M.); dreamcracker@163.com (X.S.); 2Key Laboratory of Three Gorges Regional Plant Genetics and Germplasm Enhancement (CTGU)/Biotechnology Research Center, College of Biological and Pharmaceutical Sciences, China Three Gorges University, Yichang 443002, China; 3BGI Co., Ltd. No.21 Hongan 3rd Street, Yantian District, Shenzhen 518083, China; fujie@genomics.cn

**Keywords:** sorghum bicolor, abscisic acid, saline-alkali stress

## Abstract

Sorghum (*Sorghum bicolor*) is the fifth most important cereal crop in the world. It is an annual C4 crop due to its high biomass and wide usage, and has a strong resistance to stress. Obviously, there are many benefits of planting sorghum on marginal soils such as saline-alkali land. Although it is known that abscisic acid (ABA) is involved in plant abiotic stress responses, there are few reports on sorghum. Here, we obtained RNA-seq data, which showed gene expression at the genome-wide level under saline-alkali stress. The genes related to ABA biosynthesis, catabolism, and signaling were identified and analyzed. Meanwhile, their amino acid sequences were intermingled with rice genes to form several distinct orthologous and paralogous groups. ABA-related differentially expressed genes under saline-alkali stress were identified, and family members involved in ABA signaling were hypothesized based on the expression levels and homologous genes in rice. Furthermore, the ABA signaling pathway in Sorghum bicolor was understood better by interaction analysis. These findings present a comprehensive overview of the genes regulating ABA biosynthesis, catabolism, and signaling in Sorghum bicolor under saline-alkali stress, and provide a foundation for future research regarding their biological roles in sorghum stress tolerance.

## 1. Introduction

Saline-alkaline soils are widely distributed globally, which affect seed germination, crop growth, and productivity. More than 800 million hectares of land are saline soils, and over half are alkaline [1,2]. Saline-alkaline soils are characterized by both high salinity and high alkalinity (the pH often greater than 8.0) [3]. Numerous studies have focused on the plant response mechanisms towards salt stress; less attention has been directed towards alkaline or saline-alkaline stress. It is essential to understand the responses of plants under saline-alkaline stress, in order to develop salt-alkali-tolerant crop plants. Sessile plants have evolved to possess various suitable mechanisms to adapt to abiotic stress environments. Some of the mechanisms pertaining to abiotic stress are closely related to plant hormones. Abscisic acid (ABA) is one of the major plant hormones that plays important roles in plant growth and development [4]. ABA is also called a “stress hormone” because its concentration significantly increases stress conditions and stimulates stomatal closure, gene expression, and physiological responses [5]. 

ABA is biosynthesized de novo in the plastid and cytoplasm from carotenoids [6,7]. The conversion of zeaxanthin to violaxanthin is catalyzed by zeaxanthin epoxidase (ZEP). Following this, 9-cis-epoxycarotenoids are formed by rearranging trans-violaxanthin, and then they are cleaved to produce xanthoxin by NCED (9-cis-epoxycarotenoid dioxygenase). Then in the cytoplasm, the last step of converting xanthoxin into ABA takes place, catalyzed by two enzymes, ABA2 and AAO3 [8]. ABA is irreversibly catabolized by conversion to phaseic acid (PA) through the enzyme ABA-8′-hydroxylase [9,10].

The expression of NCEDs is highly correlated with ABA levels, and the products of NCED genes are key rate-limiting enzymes in ABA biosynthesis [6]. The first NCED gene was identified from the maize vp14 mutant. Leaves from vp14 plants lose water more rapidly than wild-type maize because of reduced levels of ABA [11]. Tobacco plants transformed with NCED show enhanced drought tolerance and increased seed dormancy [12]. In Arabidopsis, five NCEDs (AtNCED2, 3, 5, 6 and 9) are most probably involved in ABA biosynthesis [13]. There are also five NCED genes implicated in ABA biosynthesis in rice. OsNCED1 has the highest expression level in leaves, and acts as the housekeeping gene under normal conditions, but it is significantly suppressed by stress conditions [14]. OsNCED2 plays a predominant role in ABA biosynthesis in germinating seeds [15]. OsNCED3 expression is highly induced by drought, salt, and hydrogen peroxide; mutant lines lacking OsNCED3 had earlier seed germination and increased sensitivity to abiotic stress, compared to wild-type lines [16]. Additionally, the expression of OsNCED4 and OsNCED5 increased under water deficiency stress [16]. The oxidation of ABA to PA is catalyzed by ABA-8′-hydroxylase, which is encoded by three genes (OsABA8ox1-3) in rice [17]. The expression of OsABAox1 was induced by cold stress, whilst the expression of OsABAox2 and OsABAox3 was not. In contrast, OsABAox2 and OsABAox3 were ABA-inducible, but OsABAox1 was not [18].

Many ABA signaling components have been identified over the past 30 years, but their relationships and a consensus on the structure of ABA signaling have been studied intensively after the identification of ABA receptors and their three-dimensional structures [19,20]. Under stress conditions, ABA accumulates in the cells and forms a complex with the ABA receptor PYR/PYL. The complex inhibits the activities of clade A protein phosphatase 2Cs (PP2Cs) allows accumulation of phosphorylated SNF1-related type 2 protein kinases (SnRK2s) and phosphorylation of the downstream substrates, such as transcription factors like bZIP and MYB [21,22,23,24]. 

The ABA receptors PYR/PYL are necessary for ABA responses. There are twelve genes encoding PYR/PYL in rice, named OsPYL1-12 [25]. He et al. found that OsPYL1-3, 6, and 10-11 display ABA-independent phosphatase inhibition manner, and OsPYL12 abrogates the phosphatase activity of PP2Cs in the absence of ABA [25]. Kim et al. demonstrated that PYR/PYLs in rice function as positive regulators of the ABA signaling pathway [26]. The rice overexpression lines of OsPYL11 (OsPYL/RCAR5) showed hypersensitivity to ABA during seed germination and early seedling growth, and improved drought and salt stress tolerance [26,27]. The overexpression of OsPYL3 and OsPYL9 substantially improved drought and cold stress tolerance in rice [28]. 

There are 78 PP2C genes in rice, and these have been divided into 11 sub-families. Bioinformatics analyses suggest that plant PP2C proteins from different sub-families participate in distinct signaling pathway [29]. Clade A PP2Cs are important components in the core ABA signaling pathway of rice. Overexpression of the clade A PP2C gene OsPP108 (OsPP2C68) enhanced resistance towards salt and drought in rice [30]. Overexpression of OsABIL2 (OsPP2C53) not only led to ABA insensitivity, but also significantly caused hypersensitivity to drought stress in rice [31]. Overexpression of OsPP2C49 in rice resulted in significantly decreased sensitivity of ABA and rapid dehydration [32].

Ten sucrose nonfermenting1 related protein kinase 2 (SnRK2) genes have been identified in the rice genome, which are designated as SAPK1-10; all members are activated by hyperosmotic stress, and SAPK8-10 are also activated by ABA [33]. The SAPK6 (OSRK1) transcript level was significantly increased by hyperosmotic stress and ABA treatment. SAPK6 can phosphorylate itself and by bZIP transcription factors such as OREB1, and ectopic expression of SAPK6 in transgenic tobacco resulted in reduced sensitivity to ABA treatment [34]. Some research has shown that SAPK4 regulates ion homeostasis, growth and development of plants under salinity stress, thereby acting as a positive regulatory factor in salt stress acclimation [35]. SAPK2 was found to interact with and phosphorylate OsbZIP23 for its transcriptional activation [32]. 

The evolution of core components of ABA signaling suggests that a basic ABA signaling pathway was already established in bryophytes, and that some components of ABA signaling are conserved [36]. It is imperative to understand ABA biosynthesis, catabolism, and signaling in crops in order to improve their tolerance towards stress. Sorghum bicolor, a C4 crop, possesses high tolerance to various abiotic stresses and is more often grown in semi-arid areas [37]. Here, we analyzed the main components of the ABA pathway such as the family members of NCEDs, PYL/PYRs, SnRKs, PP2Cs, bZIPs and ABA-8′-hydroxylase, and compared them with the family members in rice via phylogenetic analysis. An analysis of the gene expression profiles of Sorghum bicolor under different saline-alkali stress levels revealed their changes in expression level and their diverse functions. These preliminary results established the pathway of ABA biosynthesis, catabolism, and signaling in sorghum under saline-alkali stress, and presented potentially useful information for genetic improvement of crop stress tolerance.

## 2. Materials and Methods 

### 2.1. RNA-seq

Total RNA was extracted from wild type Sorghum bicolor BTx623 at the four-leaf-stage under normal conditions, moderate saline-alkali soil stress treatment (0.3% salt content, pH 8.1), severe saline-alkali soil stress treatment (0.5% salt content, pH 8.1), with three biological replicates. The RNA samples were sequenced and analyzed by Novogene Bioinformatics Technology (China) with Hiseq-PE150 (Illumina, San Diego, CA, USA). Raw sequencing reads were filtered by FASTP and mapped to the sorghum genome (v 3.1.1) using HISAT2 (version 2.0.5) with default parameters. Differentially expressed genes, which were defined with an expression change fold ≥ 2 with *p* < 0.05, were analyzed using DESeq2 (version 1.20.0).

### 2.2. Identification and Bioinformatics Analysis

The ABA-related family members in the Sorghum bicolor genome DNA were identified in the Phytozome database (https://phytozome.jgi.doe.gov/pz/portal.html). Multiple alignments were performed and phylogenetic trees were constructed by the Neighbor-joining method with 1000 replicates on each node, using TBtools [38].

### 2.3. Stress Treatments and qPCR 

The stress treatment assays were checked as described previously [39]. Briefly, the seeds were germinated on 1/2 Murashige and Skoog (MS) medium for seven days, and then transplanted to 1/2 MS medium, each containing 10 µM ABA, 200 mM NaCl, 20 mM Na_2_CO_3_ or no treatment (12 plants each and three repeats). Sampling was performed at different time points and total RNA was isolated. Total cDNA was obtained by using EasyScript One-Step gDNA Removal and cDNA Synthesis SuperMix (TransGen Biotech Co., Ltd, Beijing, China) kits. The expression levels of genes was determined by qPCR with specific primers listed in Appendix A.

### 2.4. Yeast Two-Hybrid Assay

Yeast two-hybrid (Y2H) assays were performed in the system of ProQuest two-hybrid (Invitrogen, Carlsbad, CA, USA). The full-length coding region of SbPP2C58,54.52 was fused to the GAL4 activation domain by constructing this in a pDEST22 vector. SbPYL7, SbSAPK1, and SbSAPK5 genes were introduced into the pDEST32 vector. For the interaction test, each bait construct was co-transformed in the Mav203 yeast strain, and plated on SD/-Leu-Trp-His medium. The β-galactosidase activity assay was performed, as per guidelines provided by the manufacturer (Invitrogen).

### 2.5. BiFC Assays

For the BiFC assays, the coding regions of SbPYL7 and SbSAPK5 were ligated into pVYCE and fused with the C-terminus of YFP. SbPP2C52 DNA was cloned into pVYNE and fused with the N-terminus of YFP [40]. The recombinant constructs were co-transformed in pairs into tobacco leaves, and the fluorescence signal was examined using a FluoView FV1000 confocal laser scanning microscope (Olympus Corporation, Tokyo, Japan).

## 3. Results

### 3.1. Transcriptome Profiling of Sorghum Bicolor under Saline-Alkali Stress

To elucidate possible molecular changes of *Sorghum bicolor* BTx623 under saline-alkali stress, the transcriptome was analyzed by RNA sequencing (RNA-seq) under normal conditions, moderate saline-alkali stress treatment (3% salt content, pH 8.1), and severe saline-alkali stress treatment (5% salt content, pH 8.1). After removing the adaptor, ploy -N, and low quality reads of the raw data, more than 450 million clean reads were obtained from the nine samples. An average of 96.94% of the clean data had quality scores of Q20. The percentage of total reads mapped to unique genes ranged from 92.45–94.57%. PCA was conducted with the FPKM of the nine samples, and the two-dimensional diagrams showed that samples under the same conditions clustered together, while the groups under different treatments scattered. By screening the differentially expressed genes (DEGs), the transcriptome of the treated plants exhibited significant changes compared with the controls. With the threshold of log2 fold change ≥ 2 and *p*-value ≤ 0.01, thousands of genes were identified with changes in their expression levels (Figure 1A–C). We further classified all of these genes into five groups based on their changing pattern in response to saline-alkali stress. The genes in Group A showed that the expression level significantly increased under severe saline-alkali stress, and the genes in Group B increased their expression under moderate saline-alkali stress. The genes in Group C, which we focused on, showed that the expression levels continued to rise under saline-alkali stress. The expression of genes in Group D decreased under both the saline-alkali stress conditions. The expression of genes in Group E decreased under severe saline-alkali stress conditions. It is worth mentioning that there were many ABA-related genes in Group C (Figure 1D). In order to understand the mechanisms of the ABA pathway in response to saline-alkali stress, the ABA-related genes were analyzed (Appendix A).

### 3.2. ABA Biosynthesis and Catabolism-Related Genes in Response to Saline-Alkali Stress

The SbNCED members in the *Sorghum bicolor* genome DNA were identified in the Phytozome database. Finally, five SbNCED proteins were identified and used for further studies (Figure 2A). To clarify the phylogenetic relationships among the NCED proteins in sorghum and rice, an unrooted phylogenetic tree was constructed. These proteins of rice and sorghum were intermingled to form several distinct orthologous and paralogous groups. The NCED proteins were named from their homologous proteins in rice. SbNCED1 and SbNCED2 were more closely positioned with OsNCED1 and OsNCED2 respectively, and SbNCED3 was closely placed along with OsNCED4. The protein sequence homology of SbNCED4 and SbNCED5 were low, compared with the rice NCED proteins (Figure 2B). In order to make sure these two genes belong to this family, we performed the analysis of the protein domain. The multiple sequence alignments indicate that all SbNCEDs possess conserved motif (Appendix A). Based on these results the NCED family in sorghum contains these five proteins. Meanwhile, the expression patterns of *SbNCED* genes under different saline-alkali stress were analyzed; the results showed that these genes have differential expression levels. The expression level of *SbNCED3* increased under saline-alkali stress treatment, which was similar to that seen with its homolog in rice. The expression levels of *SbNCED4* and *SbNCED5* increased under moderate or severe saline-alkali stress, respectively (Figure 2A). Consistent with *OsNCED1*, the expression of *SbNCED1* and *SbNCED2* was suppressed under saline-alkali stress treatment (Figure 2A). These results showed that *SbNCED3* could play a central role in response to saline-alkali stress.

Two genes that encode ABA-8′-hydroxylase from sorghum have been identified by phylogenetic analysis with their homologous genes in rice (Figure 3A). The results showed that there was one sorghum homolog each for rice *OsABA8OX1* and *OsABA8OX3*, namely *SbABAOX1* and *SbABAOX2* (Figure 3B). The expression pattern of a gene often has a correlation with its function, so we identified the expression profiles of these two sorghum genes form the RNA-seq results. The expression of *SbABAOX1* increased under severe saline-alkali stress treatment. The expression pattern of *SbABAOX2* displayed decreased continuously under saline-alkali stress treatment (Figure 3A). The results showed that the expression pattern of *SbABAOX* genes displayed variations in both rice and sorghum (*SbABAOX1*), and also showed similar expression profiles (*SbABAOX2*).

### 3.3. Expression of ABA Signaling-Related Genes in Response to Saline-Alkali Stress

Eleven ABA receptor candidates were identified in *Sorghum bicolor*, and a phylogenetic tree of rice and sorghum PYL/PYRs was built in order to compare genetic diversity among these proteins (Figure 4). The phylogenic tree showed that these proteins could be categorized into three subgroups (Figure 4B). Based on these results, seven sorghum PYL/PYRs proteins were closely related genetically to the PYL/PYRs proteins in rice. The PYL/PYRs proteins in sorghum were named SbPYL1-7 based on their phylogenetic relationship (Figure 4B). In order to identify the gene expression pattern in response to saline-alkali stress, a heat map analysis was performed using the data from RNA-seq experiments (Figure 4A). The sorghum ortholog of the gene encoding the rice ABA receptor OsPYL11, *SbPYL7*, was also upregulated by saline-alkali stress, which may perform a similar function to OsPYL11 in ABA signaling. 

From the RNA-seq data, 58 PP2C genes were detected (Figure 5A) and phylogenetic analyses was then conducted to divide them into sub-families, with Group A PP2C members in rice (Figure 5B). The results showed that there were ten sub-families, and there was one sorghum homolog sub-family for rice group A sub-family (Figure 5B). After analyzing the expression profiles under saline-alkali stress, it was interesting to find that the expression levels of some rice group A PP2C homologs in sorghum were increased (Figure 5A), which were named as *SbPP2C09*, *SbPP2C23, SbPP2C52, SbPP2C54,* and *SbPP2C58*, based on their location on chromosomes. Gene expression patterns can provide important clues for gene function; therefore, these up-regulated genes may play central roles in ABA signaling under saline-alkali stress.

To better understand the roles of sucrose nonfermenting1-related protein kinase2 (SnRK2) family members in response to saline-alkali stress (Figure 6A), we assembled a phylogenetic analysis of 14 SnRK2 proteins in sorghum and 10 SAPK genes in rice (Figure 6B). The sorghum SnRK2 family members were named SbSAPK1-14 based on their location on the chromosome. We found that there was no sorghum ortholog of OsSAPK3, each clade clearly contains homologs in rice from sorghum (Figure 6B). It was also notable that three SnRK2 proteins in sorghum group together and have little further genetic relationship with SAPKs in rice. Expression of the SnRK2 members was analyzed based on the RNA-seq data. The results showed that the expression of *SbSAPK1, SbSAPK5*, and *SbSAPK9* was significantly upregulated by saline-alkali stress (Figure 6A). These indicated that these genes may play vital roles in ABA signaling under saline-alkali stress.

### 3.4. Confirming the Expression Levels of ABA-Related Genes in Sorghum under Saline-Alkali Stress 

To test the accuracy of these changes in gene expression, we detected the dynamic expression levels of the ABA-related genes under 200 mM NaCl, 20 mM Na_2_CO_3_, and 10 µM ABA treatment (Figure 7). The results are mostly consistent with the RNA-seq results. The expression of *SbNCED5* was significantly upregulated under the stress treatment and reached its peak after an hour of treatment. The expression levels of *SbPYL7* peaked at one hour following saline treatment and relatively late under alkaline or ABA treatments. *SbPP2C53* and *SbPP2C58* were striking upregulated under these stress treatments, and the expression patterns were similar. Although the expression of *SbPP2C09* and *SbPP2C23* was significantly upregulated under these stress treatments, there were some different expression patterns: the expression of *SbPP2C09* peaked at one hour under alkaline treatment, and *SbPP2C23* peaked at one hour under saline treatment. *SbSAPK1* and *SbSAPK5* were up-regulated under these treatments and the patterns were similar as well. These results verified the ABA-related differential expression patterns seen in the RNA-seq experiment.

### 3.5. Identification of the Interaction between the Putative ABA Signaling Members 

Many putative ABA signaling members were identified by bioinformatics analysis, and the expression levels of some members are regulated by saline-alkaline stress. Here, the interaction between these members were detected. Full-length genes were fused with a Gal4 DNA-binding domain or activation domain; the genes included, SbPYL7, SbPP2C52, SbPP2C54, SbPP2C58, SbSAPK1, and SAPK5. In the yeast two-hybrid assay, the results showed that SbPYL7 interacted with SbPP2C52, and SbPP2C52 interacted with SbSAPK5. Simultaneously, SbPP2C54 and SbPP2C58 could interact with SbSAPK5 (Figure 8A). Following this, the interaction relationships between SbPYL7, SbPP2C52, and SbSAPK5 were confirmed using BiFC in tobacco (Figure 8B). The results showed that SbPYL7 interacted with SbPP2C52, SbPP2C52 interacted with SbSAPK5 respectively. These indicated that SbPYL7, SbPP2C52 and SbSAPK5 could form a complete ABA signaling pathway.

## 4. Discussion

Sorghum bicolor, which is the fifth largest crop in world, has high photosynthetic efficiency and environmental adaptability. Sorghum bicolor also has high tolerance to salinity, alkalinity, drought, flood, heat, and cold stress. It therefore has great potential to be planted in marginal lands such as saline-alkaline soils, to increase overall food production [41]. As the full-length genome of Sorghum bicolor (inbred line BTx623) was sequenced, many important genetic resources have been provided to further functional genomics research and genetic improvement. Here, the ABA-related genes including those involved in ABA biosynthesis, catabolism, and signaling were identified, and the expression patterns under saline-alkaline stress treatment were analyzed (Figure 1, Figure 2, Figure 3, Figure 4, Figure 5 and Figure 6). Furthermore, the interaction relationships among the ABA signaling genes were identified by yeast and BiFC assays (Figure 8).

The expression patterns of SbNCEDs and SbABAOXs showed that SbNCED5 was induced more strongly than SbABAOX1 under saline-alkaline stress, indicating that ABA may be synthesized in large quantities, especially at the early stages of stress treatment (Figure 2 and Figure 3). In this study, 7 PYL/PCAR genes were identified and only four members changed their expression under saline-alkaline stress (Figure 4). It was reported that group C of PP2C family plays a role in ABA signaling [42]. There are 20 genes in group C of sorghum, and five of them were upregulated under saline-alkaline stress in this study (Figure 5). Furthermore, two members of 14 SAPKs were obviously upregulated under the stress (Figure 6). These results indicated that there are specific members involved in ABA biosynthesis, catabolism, and signaling. It is urgent to make further study of these specific functions of these genes under saline-alkali stress, and it is better for exploring the regulatory function of ABA in saline-alkali stress.

In the interaction assay, we cloned the full-length genes for further studies (Figure 8). An ABA signaling pathway was constructed with SbPYL7, SbPP2C52 and SbSAPK5 under saline-alkaline stress. It is worth mentioning that the SbPYL7 interacted with SbPP2C52 independently of ABA. In rice and *Arabidopsis*, it has been reported that PYL interact with clade A PP2Cs may be in an ABA-dependent fashion using the Y2H assays [26]. So it is necessary to detect more interactions by adding ABA into the medium in the further studies. The results indicated that ABA plays a vital role in response to saline-alkaline stress and specific members take part in ABA signaling. In future studies, based on the ABA signaling analysis we have conducted, it is necessary to find the downstream members and their regulators, in order to reveal the mechanisms by which sorghum responds to saline-alkaline stress, thereby providing more information for crop genetic improvement.

## 5. Conclusions

Our study has identified the genes of abscisic acid biosynthesis, catabolism, and signaling in Sorghum bicolor under saline-alkali stress by RNA-seq data and bioinformatics analysis, and the ABA signaling pathway were understood and established well by interaction analysis.

## Figures and Tables

**Figure 1 biomolecules-09-00823-f001:**
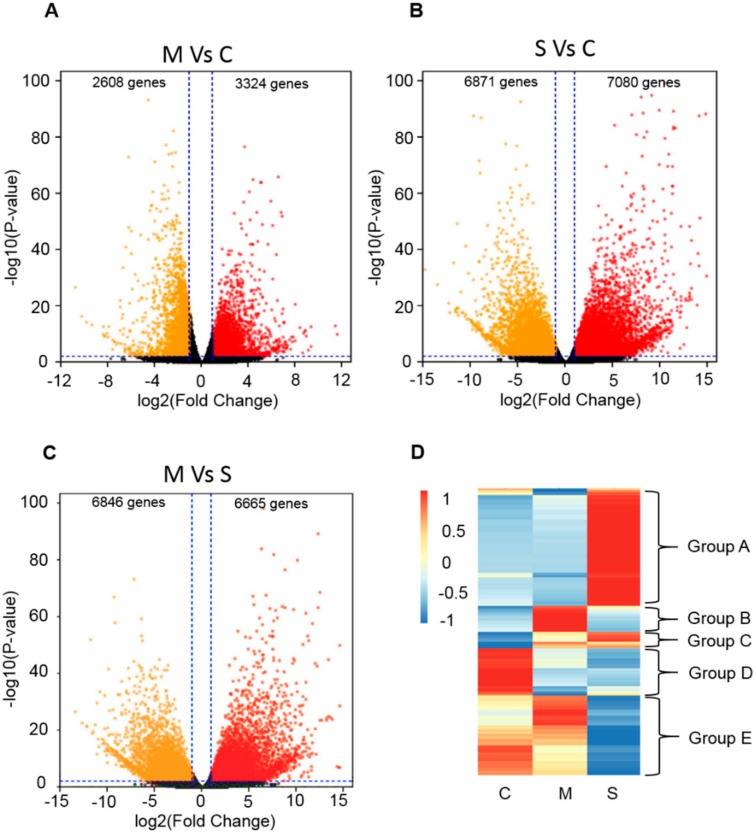
Analysis of differentially expressed genes under saline-alkali stress. (**A**–**C**), Volcano plots of differentially expressed genes (DEGs) among normal condition (**C**), moderate (M) and severe (S) saline-alkali stress in sorghum bicolor: (**A**)moderate stress treatment vs. control; (**B**) severe stress treatment vs. control; (**C**) moderate stress treatment severe stress treatment. (**D**), Expression patterns of differentially expressed genes (DEGs) among normal condition (**C**), moderate (M) and severe (S) saline-alkali stress in sorghum bicolor.

**Figure 2 biomolecules-09-00823-f002:**
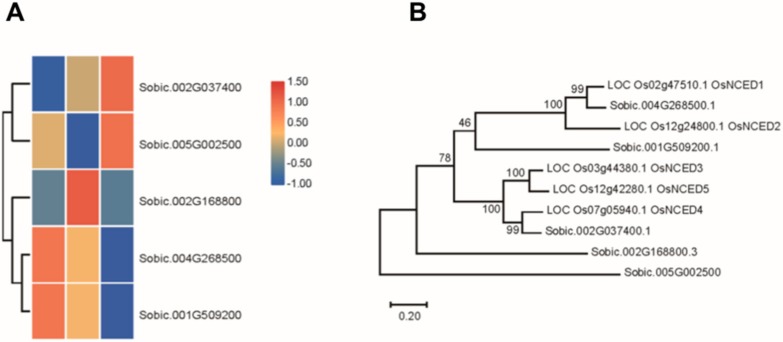
Analysis of *NCED* genes in sorghum. (**A**) The expression levels of *NCED* genes in sorghum. (**B**) Evolutionary analysis of SbNCED proteins and rice OsNCED proteins.

**Figure 3 biomolecules-09-00823-f003:**
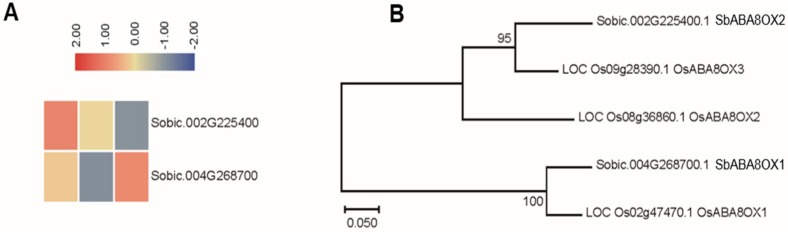
Analysis of the ABA-8′-hydroxylase encoded genes in sorghum. (**A**) The expression patterns of ABA8OX genes in sorghum. (**B**) Phylogenetic relationship of SbABA8OX proteins and rice OsABA8OX proteins.

**Figure 4 biomolecules-09-00823-f004:**
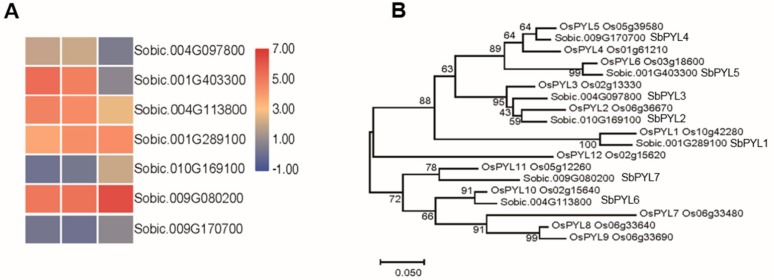
Analysis of the ABA receptors in sorghum. (**A**) The expression patterns of PYLs in sorghum. (**B**) The phylogenetic tree constructed by PYLs in sorghum and rice.

**Figure 5 biomolecules-09-00823-f005:**
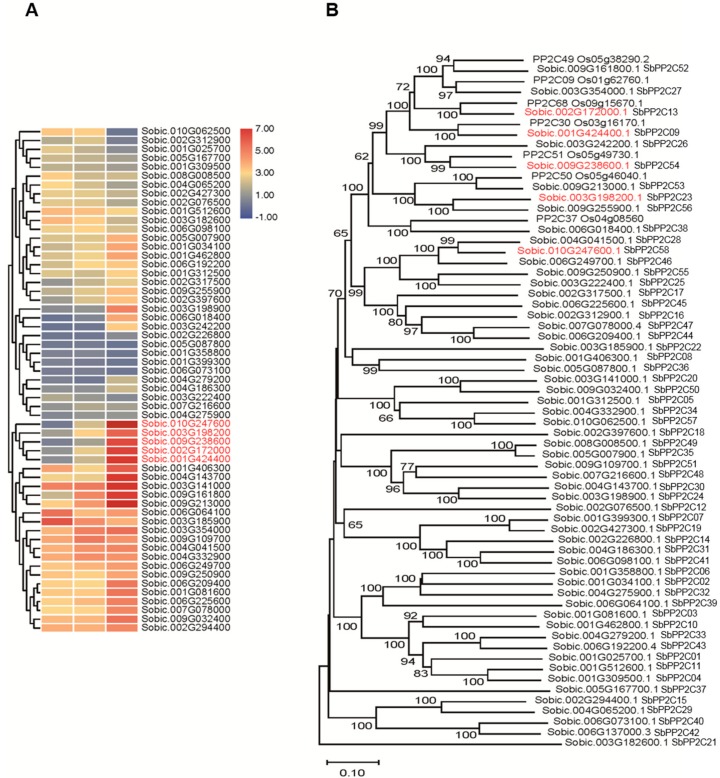
Analysis of the *PP2Cs* genes in sorghum. (**A**) The expression patterns of *PP2Cs* in sorghum. (**B**) The phylogenetic relationship of SbPP2Cs and group A OsPP2Cs.

**Figure 6 biomolecules-09-00823-f006:**
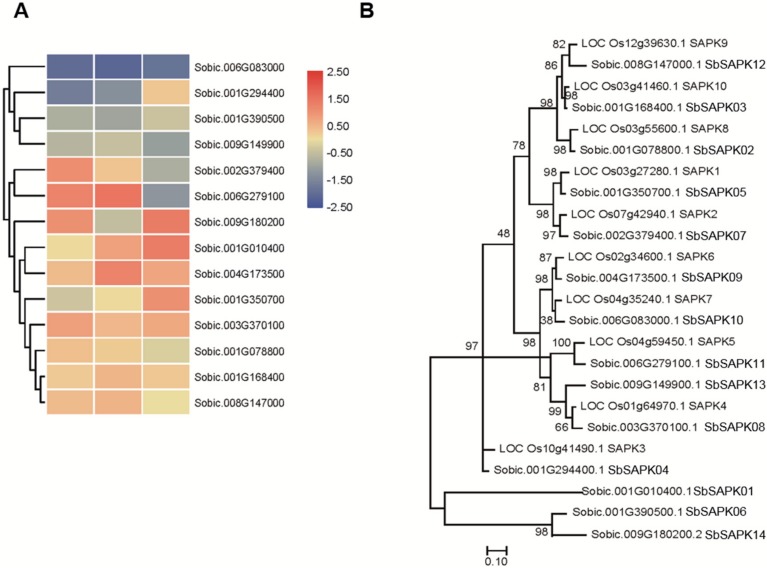
Analysis of *SbSAPK* genes in sorghum. (**A**) The expression patterns of SbSAPKs in sorghum under saline-alkali stress. (**B**) The phylogenetic relationship of SAPKs in sorghum and rice.

**Figure 7 biomolecules-09-00823-f007:**
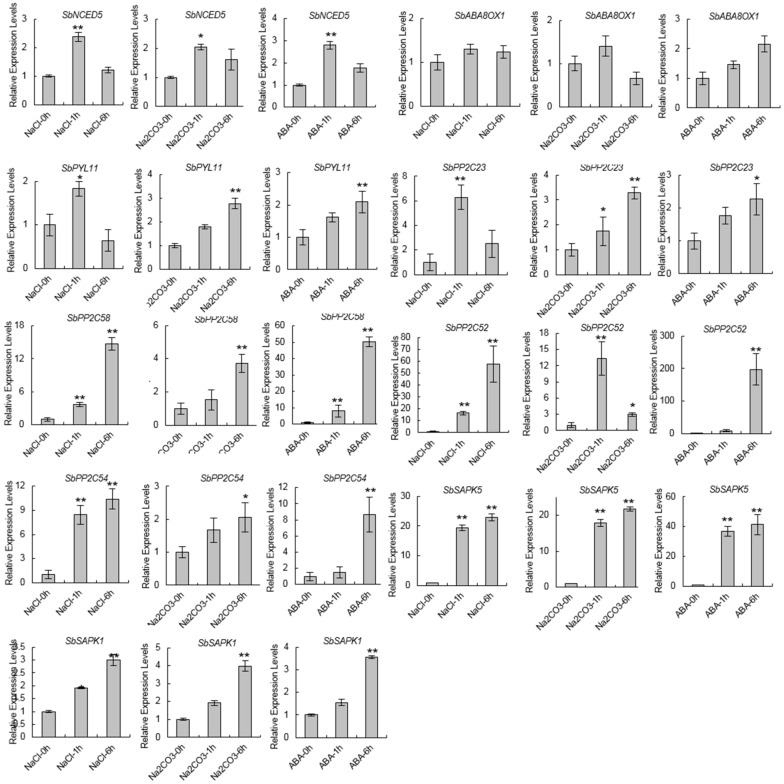
The relative expression levels of ABA-related genes under stress treatment. The x-axis indicates the time course of the sampling (h, hours). Error bars represent the standard error (SE) based on three biological replicates. The statistical significance was determined by Student’s *t*-test (** *p* < 0.01, * *p* < 0.05, two-sided).

**Figure 8 biomolecules-09-00823-f008:**
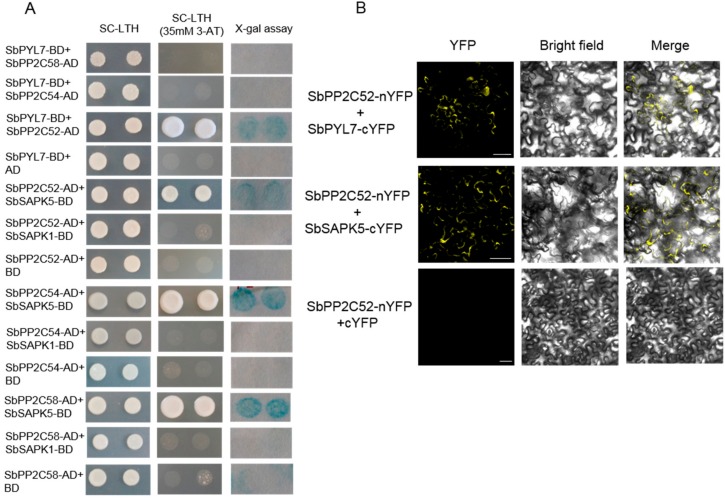
The interaction relationships of ABA-related proteins. (**A**) Interaction between these proteins detected by yeast-two hybrid assay. SC-LTH, synthetic complete Leu-Trp-His medium; 3-AT, 3-amino-1,2,4-triazloe; AD, GAL4 activation domain. BD, GAL4 DNA-binding domain. (**B**) Confirmation of the interaction by BiFC assay in tobacco. Bars = 20 µm.

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
