# Peer review of "Genome-Wide Analysis of Abscisic Acid Biosynthesis, Catabolism, and Signaling in Sorghum Bicolor under Saline-Alkali Stress"

_biomolecules, 2019, doi:10.3390/biom9120823_

Round 1
Reviewer 1 Report
Article is written well with an acceptable novelty, but here are three questions which should be answered by the authors.
1) Line 205: How the authors get this fact about the conserved expression profiles while they are comparing only two different species?
2) Figure 7: Three biological replicates or three technical repeats?
3) Figure 7: Regarding with the comparing of the meaning expression levels from three biological (or three technical), why the authors are presenting SD, and not SE?
Reviewer 2 Report
This report does not provide new iinsight for the study of ABA signaling, but is useful in providing information that will serve as the basis for conducting on sorghum ABA signaling research.To improve the manuscript, my specific comments are listed below.
1. In phylogenetic tree of NCEDs, sorghum-specific clusters (SbNCED4, SbNCED5) are formed. In order to determine whether these proteins are as NCED, it will be necessary to provide an appropriate out group for drawing the phylogenetic tree. The same applies to SbPYL6,7,8,9.
2. Description of accession number of newly identified gene and NGS data cannot be confirmed. In general, it is strongly recommended to register gene sequences and ngs data in public databases when determining new gene sequences or used NGS data for research. If you have not registered yet, please register and describe the registration number in the paper.
3. When assaying the interaction between Group A PP2Cs and PYLs by Y2H, there are combinations that do not interact unless ABA is added to the medium. Was it considered ?
Round 2
Reviewer 1 Report
The authors have tried to address comments well. But here we have some new problems, which must be revised before cosidering to publish.
Major:
1- Figure 2C: How readers are able to figure the similarity of the mentioned five proteins out from this schematic? The authors must reveal the allignments of proteins as the supplementary and also compare the protein structures to show the exact similarities instead of Figure 2C.
Minor:
Line 188: "really belong" is not scientific. Please rewrite.
Line 269: Standard is duplicated.
Author Response
The authors have tried to address comments well. But here we have some new problems, which must be revised before cosidering to publish.
Major:
1- Figure 2C: How readers are able to figure the similarity of the mentioned five proteins out from this schematic? The authors must reveal the allignments of proteins as the supplementary and also compare the protein structures to show the exact similarities instead of Figure 2C.
Response:Thanks for your comments,we made the amino acid sequences aligments and motif distribution analysis in Figure S1.
Minor:
Line 188: "really belong" is not scientific. Please rewrite.
Response: Thanks. we have rewrited it in line 188.
Line 269: Standard is duplicated.
Response: Thanks. We have deleted it.